# Chain-of-Thought Prompting Elicits Reasoning in Large Language Models

**Jason Wei**    **Xuezhi Wang**    **Dale Schuurmans**    **Maarten Bosma**

**Brian Ichter**    **Fei Xia**    **Ed H. Chi**    **Quoc V. Le**    **Denny Zhou**

Google Research, Brain Team
{jasonwei,dennyzhou}@google.com

## Abstract

We explore how generating a *chain of thought*—a series of intermediate reasoning steps—significantly improves the ability of large language models to perform complex reasoning. In particular, we show how such reasoning abilities emerge naturally in sufficiently large language models via a simple method called *chain-of-thought prompting*, where a few chain of thought demonstrations are provided as exemplars in prompting.

Experiments on three large language models show that chain-of-thought prompting improves performance on a range of arithmetic, commonsense, and symbolic reasoning tasks. The empirical gains can be striking. For instance, prompting a PaLM 540B with just eight chain-of-thought exemplars achieves state-of-the-art accuracy on the GSM8K benchmark of math word problems, surpassing even finetuned GPT-3 with a verifier.

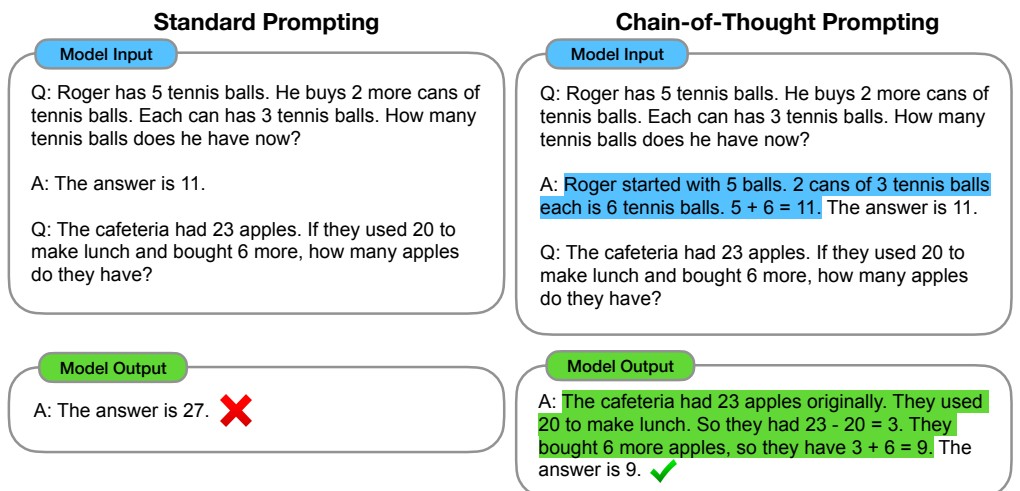

Figure 1: Chain-of-thought prompting enables large language models to tackle complex arithmetic, commonsense, and symbolic reasoning tasks. Chain-of-thought reasoning processes are highlighted.

36th Conference on Neural Information Processing Systems (NeurIPS 2022).

# 1 Introduction

The NLP landscape has recently been revolutionized by language models (Peters et al., 2018; Devlin et al., 2019; Brown et al., 2020, *inter alia*). Scaling up the size of language models has been shown to confer a range of benefits, such as improved performance and sample efficiency (Kaplan et al., 2020; Brown et al., 2020, *inter alia*). However, scaling up model size alone has not proved sufficient for achieving high performance on challenging tasks such as arithmetic, commonsense, and symbolic reasoning (Rae et al., 2021).

This work explores how the reasoning ability of large language models can be unlocked by a simple method motivated by two ideas. First, techniques for arithmetic reasoning can benefit from generating natural language rationales that lead to the final answer. Prior work has given models the ability to generate natural language intermediate steps by training from scratch (Ling et al., 2017) or finetuning a pretrained model (Cobbe et al., 2021), in addition to neuro-symbolic methods that use formal languages instead of natural language (Roy and Roth, 2015; Chiang and Chen, 2019; Amini et al., 2019; Chen et al., 2019). Second, large language models offer the exciting prospect of in-context few-shot learning via *prompting*. That is, instead of finetuning a separate language model checkpoint for each new task, one can simply "prompt" the model with a few input–output exemplars demonstrating the task. Remarkably, this has been successful for a range of simple question-answering tasks (Brown et al., 2020).

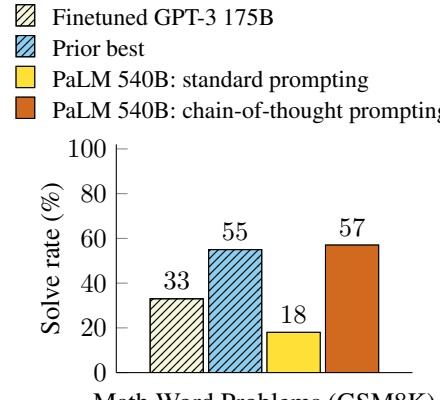

Figure 2: PaLM 540B uses chain-of-thought prompting to achieve new state-of-the-art performance on the GSM8K benchmark of math word problems. Finetuned GPT-3 and prior best are from Cobbe et al. (2021).

Both of the above ideas, however, have key limitations. For rationale-augmented training and finetuning methods, it is costly to create a large set of high quality rationales, which is much more complicated than simple input–output pairs used in normal machine learning. For the traditional few-shot prompting method used in Brown et al. (2020), it works poorly on tasks that require reasoning abilities, and often does not improve substantially with increasing language model scale (Rae et al., 2021). In this paper, we combine the strengths of these two ideas in a way that avoids their limitations. Specifically, we explore the ability of language models to perform few-shot prompting for reasoning tasks, given a prompt that consists of triples: ⟨input, *chain of thought*, output⟩. A *chain of thought* is a series of intermediate natural language reasoning steps that lead to the final output, and we refer to this approach as *chain-of-thought prompting*. An example prompt is shown in Figure 1.

We present empirical evaluations on arithmetic, commonsense, and symbolic reasoning benchmarks, showing that chain-of-thought prompting outperforms standard prompting, sometimes to a striking degree. Figure 2 illustrates one such result—on the GSM8K benchmark of math word problems (Cobbe et al., 2021), chain-of-thought prompting with PaLM 540B outperforms standard prompting by a large margin and achieves new state-of-the-art performance. A prompting only approach is important because it does not require a large training dataset and because a single model checkpoint can perform many tasks without loss of generality. This work underscores how large language models can learn via a few examples with natural language data about the task (c.f. automatically learning the patterns underlying inputs and outputs via a large training dataset).

# 2 Chain-of-Thought Prompting

Consider one's own thought process when solving a complicated reasoning task such as a multi-step math word problem. It is typical to decompose the problem into intermediate steps and solve each before giving the final answer: *"After Jane gives 2 flowers to her mom she has 10 . . . then after she gives 3 to her dad she will have 7 . . . so the answer is 7."* The goal of this paper is to endow language models with the ability to generate a similar *chain of thought*—a coherent series of intermediate reasoning steps that lead to the final answer for a problem. We will show that sufficiently large

language models can generate chains of thought if demonstrations of chain-of-thought reasoning are provided in the exemplars for few-shot prompting.

Figure 1 shows an example of a model producing a chain of thought to solve a math word problem that it would have otherwise gotten incorrect. The chain of thought in this case resembles a solution and can interpreted as one, but we still opt to call it a chain of thought to better capture the idea that it mimics a step-by-step thought process for arriving at the answer (and also, solutions/explanations typically come *after* the final answer (Narang et al., 2020; Wiegreffe et al., 2022; Lampinen et al., 2022, *inter alia*)).

Chain-of-thought prompting has several attractive properties as an approach for facilitating reasoning in language models.

1. First, chain of thought, in principle, allows models to decompose multi-step problems into intermediate steps, which means that additional computation can be allocated to problems that require more reasoning steps.

2. Second, a chain of thought provides an interpretable window into the behavior of the model, suggesting how it might have arrived at a particular answer and providing opportunities to debug where the reasoning path went wrong (although fully characterizing a model's computations that support an answer remains an open question).

3. Third, chain-of-thought reasoning can be used for tasks such as math word problems, commonsense reasoning, and symbolic manipulation, and is potentially applicable (at least in principle) to any task that humans can solve via language.

4. Finally, chain-of-thought reasoning can be readily elicited in sufficiently large off-the-shelf language models simply by including examples of chain of thought sequences into the exemplars of few-shot prompting.

In empirical experiments, we will observe the utility of chain-of-thought prompting for arithmetic reasoning (Section 3), commonsense reasoning (Section 4), and symbolic reasoning (Section 5).

## 3 Arithmetic Reasoning

We begin by considering math word problems of the form in Figure 1, which measure the arithmetic reasoning ability of language models. Though simple for humans, arithmetic reasoning is a task where language models often struggle (Hendrycks et al., 2021; Patel et al., 2021, *inter alia*). Strikingly, chain-of-thought prompting when used with the 540B parameter language model performs comparably with task-specific finetuned models on several tasks, even achieving new state of the art on the challenging GSM8K benchmark (Cobbe et al., 2021).

### 3.1 Experimental Setup

We explore chain-of-thought prompting for various language models on multiple benchmarks.

**Benchmarks.** We consider the following five math word problem benchmarks: **(1)** the **GSM8K** benchmark of math word problems (Cobbe et al., 2021), **(2)** the **SVAMP** dataset of math word problems with varying structures (Patel et al., 2021), **(3)** the **ASDiv** dataset of diverse math word problems (Miao et al., 2020), **(4)** the **AQuA** dataset of algebraic word problems, and **(5)** the **MAWPS** benchmark (Koncel-Kedziorski et al., 2016). Example problems are given in Appendix Table 12.

**Standard prompting.** For the baseline, we consider standard few-shot prompting, popularized by Brown et al. (2020), in which a language model is given in-context exemplars of input–output pairs before outputting a prediction for a test-time example. Exemplars are formatted as questions and answers. The model gives the answer directly, as shown in Figure 1 (left).

**Chain-of-thought prompting.** Our proposed approach is to augment each exemplar in few-shot prompting with a chain of thought for an associated answer, as illustrated in Figure 1 (right). As most of the datasets only have an evaluation split, we manually composed a set of eight few-shot exemplars with chains of thought for prompting—Figure 1 (right) shows one chain of thought exemplar, and the full set of exemplars is given in Appendix Table 20. (These particular exemplars did not undergo prompt engineering; robustness is studied in Section 3.4 and Appendix A.2.) To investigate whether chain-of-thought prompting in this form can successfully elicit successful reasoning across a range of

**Math Word Problems (free response)**

Q: Roger has 5 tennis balls. He buys 2 more cans of tennis balls. Each can has 3 tennis balls. How many tennis balls does he have now?

A: Roger started with 5 balls. 2 cans of 3 tennis balls each is 6 tennis balls. 5 + 6 = 11. The answer is 11.

**Math Word Problems (multiple choice)**

Q: How many keystrokes are needed to type the numbers from 1 to 500?
Answer Choices: (a) 1156 (b) 1392 (c) 1480 (d) 1562 (e) 1788

A: There are 9 one-digit numbers from 1 to 9. There are 90 two-digit numbers from 10 to 99. There are 401 three-digit numbers from 100 to 500. 9 + 90(2) + 401(3) = 1392. The answer is (b).

**CSQA (commonsense)**

Q: Sammy wanted to go to where the people were. Where might he go?
Options: (a) race track (b) populated areas (c) desert (d) apartment (e) roadblock

A: The answer must be a place with a lot of people. Race tracks, desert, apartments, and roadblocks don't have a lot of people, but populated areas do. So the answer is (b).

**StrategyQA**

Q: Yes or no: Would a pear sink in water?

A: The density of a pear is about 0.6 g/cm^3, which is less than water. Thus, a pear would float. So the answer is no.

**Date Understanding**

Q: The concert was scheduled to be on 06/01/1943, but was delayed by one day to today. What is the date 10 days ago in MM/DD/YYYY?

A: One day after 06/01/1943 is 06/02/1943, so today is 06/02/1943. 10 days before today is 05/23/1943. So the answer is 05/23/1943.

**Sports Understanding**

Q: Is the following sentence plausible? "Joao Moutinho caught the screen pass in the NFC championship."

A: Joao Moutinho is a soccer player. The NFC championship is part of American football, not soccer. So the answer is no.

**SayCan (Instructing a robot)**

Human: How would you bring me something that isn't a fruit?

Explanation: the user wants something to eat that isn't a fruit. An energy bar is not a fruit, so I will bring the user an energy bar. Plan: 1. find(energy bar) 2. pick(energy bar) 3. find(user) 4. put(energy bar) 5. done().

**Last Letter Concatenation**

Q: Take the last letters of the words in "Lady Gaga" and concatenate them.

A: The last letter of "Lady" is "y". The last letter of "Gaga" is "a". Concatenating them is "ya". So the answer is ya.

**Coin Flip (state tracking)**

Q: A coin is heads up. Maybelle flips the coin. Shalonda does not flip the coin. Is the coin still heads up?

A: The coin was flipped by Maybelle. So the coin was flipped 1 time, which is an odd number. The coin started heads up, so after an odd number of flips, it will be tails up. So the answer is no.

Figure 3: Examples of ⟨input, chain of thought, output⟩ triples for arithmetic, commonsense, and symbolic reasoning benchmarks. Chains of thought are highlighted. Full prompts in Appendix G.

math word problems, we used this single set of eight chain of thought exemplars for all benchmarks except AQuA, which is multiple choice instead of free response. For AQuA, we used four exemplars and solutions from the training set, as given in Appendix Table 21.

**Language models.** We evaluate five large language models. The first is **GPT-3** (Brown et al., 2020), for which we use text-ada-001, text-babbage-001, text-curie-001, and text-davinci-002, which presumably correspond to InstructGPT models of 350M, 1.3B, 6.7B, and 175B parameters (Ouyang et al., 2022).The second is **LaMDA** (Thoppilan et al., 2022), which has models of 422M, 2B, 8B, 68B, and 137B parameters. The third is **PaLM**, which has models of 8B, 62B, and 540B parameters. The fourth is **UL2 20B** (Tay et al., 2022), and the fifth is **Codex** (Chen et al., 2021, code-davinci-002 in the OpenAI API). We sample from the models via greedy decoding (though follow-up work shows chain-of-thought prompting can be improved by taking the majority final answer over many sampled generations (Wang et al., 2022a)). For LaMDA, we report averaged results over five random seeds, where each seed had a different randomly shuffled order of exemplars. As LaMDA experiments did not show large variance among different seeds, to save compute we report results for a single exemplar order for all other models.

## 3.2 Results

The strongest results of chain-of-thought prompting are summarized in Figure 4, with all experimental outputs for each model collection, model size, and benchmark shown in Table 2 in the Appendix. There are three key takeaways. First, Figure 4 shows that chain-of-thought prompting is an emergent ability of model scale (Wei et al., 2022b). That is, chain-of-thought prompting does not positively impact performance for small models, and only yields performance gains when used with models of ∼100B parameters. We qualitatively found that models of smaller scale produced fluent but illogical chains of thought, leading to lower performance than standard prompting.

Second, chain-of-thought prompting has larger performance gains for more-complicated problems. For instance, for GSM8K (the dataset with the lowest baseline performance), performance more than doubled for the largest GPT and PaLM models. On the other hand, for SingleOp, the easiest subset of MAWPS which only requires a single step to solve, performance improvements were either negative or very small (see Appendix Table 3).

Third, chain-of-thought prompting via GPT-3 175B and PaLM 540B compares favorably to prior state of the art, which typically finetunes a task-specific model on a labeled training dataset. Figure 4 shows how PaLM 540B uses chain-of-thought prompting to achieve new state of the art on GSM8K, SVAMP, and MAWPS (though note that standard prompting already passed the prior best for SVAMP). On the other two datasets, AQuA and ASDiv, PaLM with chain-of-thought prompting reaches within 2% of the state of the art (Appendix Table 2).

To better understand why chain-of-thought prompting works, we manually examined model-generated chains of thought by LaMDA 137B for GSM8K. Of 50 random examples where the model returned the correct final answer, all of the generated chains of thought were also logically and mathematically correct except two that coincidentally arrived at the correct answer (see Appendix D.1, and Table 8 for examples of correct model-generated chains of thought). We also randomly examined 50 random samples for which the model gave the wrong answer. The summary of this analysis is that 46% of the chains of thought were almost correct, barring minor mistakes (calculator error, symbol mapping error, or one reasoning step missing), and that the other 54% of the chains of thought had major

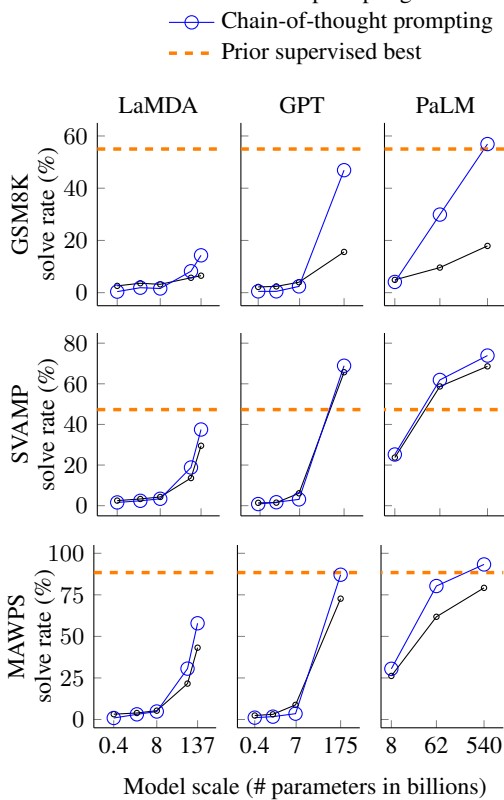

Figure 4: Chain-of-thought prompting enables large language models to solve challenging math problems. Notably, chain-of-thought reasoning is an emergent ability of increasing model scale. Prior best numbers are from Cobbe et al. (2021) for GSM8K, Jie et al. (2022) for SVAMP, and Lan et al. (2021) for MAWPS.

errors in semantic understanding or coherence (see Appendix D.2). To provide a small insight into why scaling improves chain-of-thought reasoning ability, we performed a similar analysis of errors made by PaLM 62B and whether those errors were fixed by scaling to PaLM 540B. The summary is that scaling PaLM to 540B fixes a large portion of one-step missing and semantic understanding errors in the 62B model (see Appendix A.1).

### 3.3 Ablation Study

The observed benefits of using chain-of-thought prompting raises the natural question of whether the same performance improvements can be conferred via other types of prompting. Figure 5 shows an ablation study with three variations of chain of thought described below.

**Equation only.** One reason for why chain-of-thought prompting might help is that it produces the mathematical equation to be evaluated, and so we test a variation where the model is prompted to output only a mathematical equation before giving the answer. Figure 5 shows that equation only prompting does not help much for GSM8K, which implies that the semantics of the questions in GSM8K are too challenging to directly translate into an equation without the natural language reasoning steps in chain of thought. For datasets of one-step or two-step problems, however, we find that equation only prompting does improve performance, since the equation can be easily derived from the question (see Appendix Table 6).

**Variable compute only.** Another intuition is that chain of thought allows the model to spend more computation (i.e., intermediate tokens) on harder problems. To isolate the effect of variable computation from chain-of-thought reasoning, we test a configuration where the model is prompted to output a only sequence of dots (. . .) equal to the number of characters in the equation needed to solve the problem. This variant performs about the same as the baseline, which suggests that variable computation by itself is not the reason for the success of chain-of-thought prompting, and that there appears to be utility from expressing intermediate steps via natural language.

**Chain of thought after answer.** Another potential benefit of chain-of-thought prompting could simply be that such prompts allow the model to better access relevant knowledge acquired during pretraining. Therefore, we test an alternative configuration where the chain of thought prompt is only given after the answer, isolating whether the model actually depends on the produced chain of thought to give the final answer. This variant performs about the same as the baseline, which suggests that the sequential reasoning embodied in the chain of thought is useful for reasons beyond just activating knowledge.

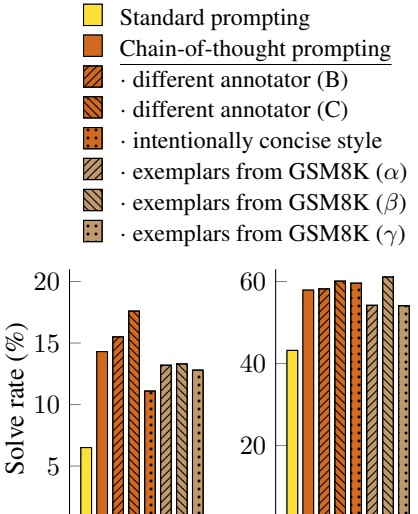

Figure 5: Ablation study for different variations of prompting using LaMDA 137B and PaLM 540B. Results for other datasets are given in Appendix Table 6 and Table 7.

## 3.4 Robustness of Chain of Thought

Sensitivity to exemplars is a key consideration of prompting approaches—for instance, varying the permutation of few-shot exemplars can cause the accuracy of GPT-3 on SST-2 to range from near chance (54.3%) to near state of the art (93.4%) (Zhao et al., 2021). In this final subsection, we evaluate robustness to chains of thought written by different annotators. In addition to the results above, which used chains of thought written by an Annotator A, two other co-authors of this paper (Annotators B and C) independently wrote chains of thought for the same few-shot exemplars (shown in Appendix H). Annotator A also wrote another chain of thought that was more concise than the original, following the style of solutions given in Cobbe et al. (2021).[1]

Figure 6 shows these results for LaMDA 137B on GSM8K and MAWPS (ablation results for other datasets are given in Appendix Table 6 / Table 7). Although there is variance among different chain of thought annotations, as would be expected when using exemplar-based prompting (Le Scao and Rush, 2021; Reynolds and McDonell, 2021; Zhao et al., 2021), all sets of chain of thought prompts outperform the standard baseline by a large margin. This result implies that successful use of chain of thought does not depend on a particular linguistic style.

To confirm that successful chain-of-thought prompting works for other sets of exemplars, we also run experiments with three sets of eight exemplars randomly sampled from the GSM8K training set, an independent

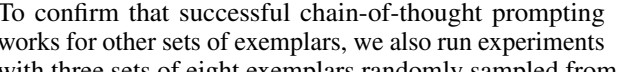

Figure 6: Chain-of-thought prompting has variance for different prompt examples (as expected) but outperforms standard prompting for various annotators as well as for different exemplars.

---

[1]For instance, whereas original chain of thought uses several short sentences (*"'There were originally 9 computers. For each of 4 days, 5 more computers were added. So 5 * 4 = 20 computers were added. 9 + 20 is 29."*), the concise chain of thought would read *"5 * 4 = 20 new computers were added. So there are 9 + 20 = 29 new computers in the server room now"*.

source (examples in this dataset already included reasoning steps like a chain of thought).[2] Figure 6 shows that these prompts performed comparably with our manually written exemplars, also substantially outperforming standard prompting.

In addition to robustness to annotators, independently-written chains of thought, different exemplars, and various language models, we also find that chain-of-thought prompting for arithmetic reasoning is robust to different exemplar orders and varying numbers of exemplars (see Appendix A.2).

# 4   Commonsense Reasoning

Although chain of thought is particularly suitable for math word problems, the language-based nature of chain of thought actually makes it applicable to a broad class of commonsense reasoning problems, which involve reasoning about physical and human interactions under the presumption of general background knowledge. Commonsense reasoning is key for interacting with the world and is still beyond the reach of current natural language understanding systems (Talmor et al., 2021).

**Benchmarks.** We consider five datasets covering a diverse range of commonsense reasoning types. The popular **CSQA** (Talmor et al., 2019) asks commonsense questions about the world involving complex semantics that often require prior knowledge. **StrategyQA** (Geva et al., 2021) requires models to infer a multi-hop strategy to answer questions. We choose two specialized evaluation sets from the BIG-bench effort (BIG-bench collaboration, 2021): **Date** Understanding, which involves inferring a date from a given context, and **Sports** Understanding, which involves determining whether a sentence relating to sports is plausible or implausible. Finally, the **SayCan** dataset (Ahn et al., 2022) involves mapping a natural language instruction to a sequence of robot actions from a discrete set. Figure 3 shows examples with chain of thought annotations for all datasets.

**Prompts.** We follow the same experimental setup as the prior section. For CSQA and StrategyQA, we randomly selected examples from the training set and manually composed chains of thought for them to use as few-shot exemplars. The two BIG-bench tasks do not have training sets, so we selected the first ten examples as exemplars in the evaluation set as few-shot exemplars and report numbers on the rest of the evaluation set. For SayCan, we use six examples from the training set used in Ahn et al. (2022) and also manually composed chains of thought.

**Results.** Figure 7 highlights these results for PaLM (full results for LaMDA, GPT-3, and different model scales are shown in Table 4). For all tasks, scaling up model size improved the performance of standard prompting; chain-of-thought prompting led to further gains, with improvements appearing to be largest for PaLM 540B. With chain-of-thought prompting, PaLM 540B achieved strong performance relative to baselines, outperforming the prior state of the art on StrategyQA (75.6% vs 69.4%) and outperforming an unaided sports enthusiast on sports understanding (95.4% vs 84%). These results demonstrate that chain-of-thought prompting can also improve performance on tasks requiring a range of commonsense reasoning abilities (though note that gain was minimal on CSQA).

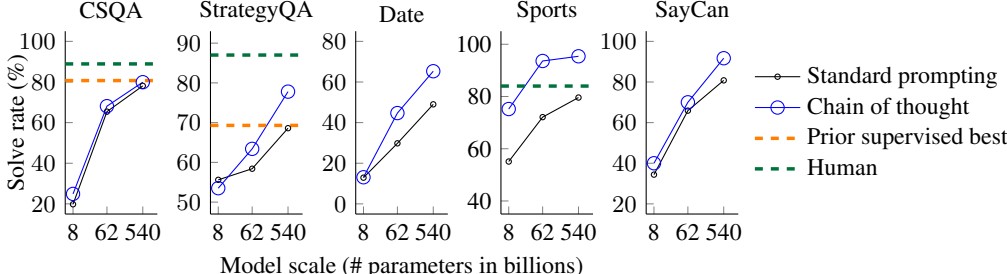

Figure 7:   Chain-of-thought prompting also improves the commonsense reasoning abilities of language models. The language model shown here is PaLM. Prior best numbers are from the leaderboards of CSQA (Talmor et al., 2019) and StrategyQA (Geva et al., 2021) (single-model only, as of May 5, 2022). Additional results using various sizes of LaMDA, GPT-3, and PaLM are shown in Table 4.

---

[2] We sample examples $\leq 60$ tokens to fit into our input context window, and also limit the examples to $\leq 2$ steps to solve for a fair comparison with the eight exemplars that we composed.

# 5 Symbolic Reasoning

Our final experimental evaluation considers symbolic reasoning, which is simple for humans but potentially challenging for language models. We show that chain-of-thought prompting not only enables language models to perform symbolic reasoning tasks that are challenging in the standard prompting setting, but also facilitates length generalization to inference-time inputs longer than those seen in the few-shot exemplars.

**Tasks.** We use the following two toy tasks.

- **Last letter concatenation.** This task asks the model to concatenate the last letters of words in a name (e.g., *"Amy Brown"* → *"yn"*). It is a more challenging version of first letter concatenation, which language models can already perform without chain of thought.[3] We generate full names by randomly concatenating names from the top one-thousand first and last names from name census data (`https://namecensus.com/`).
- **Coin flip.** This task asks the model to answer whether a coin is still heads up after people either flip or don't flip the coin (e.g., *"A coin is heads up. Phoebe flips the coin. Osvaldo does not flip the coin. Is the coin still heads up?"* → *"no"*).

As the construction of these symbolic reasoning tasks is well-defined, for each task we consider an *in-domain* test set for which examples had the same number of steps as

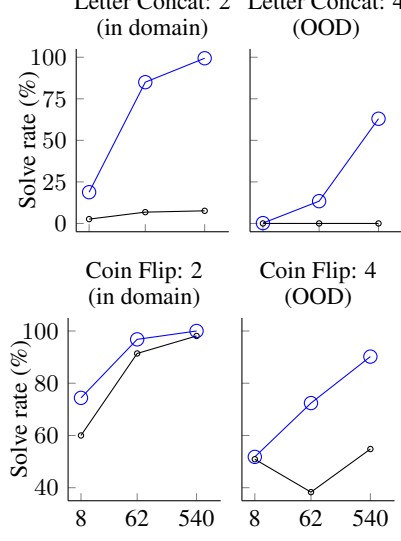

Figure 8: Using chain-of-thought prompting facilitates generalization to longer sequences in two symbolic reasoning tasks.

the training/few-shot exemplars, as well as an *out-of-domain* (OOD) test set, for which evaluation examples had more steps than those in the exemplars. For last letter concatenation, the model only sees exemplars of names with two words, and then performs last letter concatenation on names with 3 and 4 words.[4] We do the same for the number of potential flips in the coin flip task. Our experimental setup uses the same methods and models as in the prior two sections. We again manually compose chains of thought for the few-shot exemplars for each task, which are given in Figure 3.

**Results.** The results of these in-domain and OOD evaluations are shown in Figure 8 for PaLM, with results for LaMDA shown in Appendix Table 5. With PaLM 540B, chain-of-thought prompting leads to almost 100% solve rates (note that standard prompting already solves coin flip with PaLM 540, though not for LaMDA 137B). Note that these in-domain evaluations are "toy tasks" in the sense that perfect solution structures are already provided by the chains of thought in the few-shot exemplars; all the model has to do is repeat the same steps with the new symbols in the test-time example. And yet, small models still fail—the ability to perform abstract manipulations on unseen symbols for these three tasks only arises at the scale of 100B model parameters.

As for the OOD evaluations, standard prompting fails for both tasks. With chain-of-thought prompting, language models achieve upward scaling curves (though performance is lower than in the in-domain setting). Hence, chain-of-thought prompting facilitates length generalization beyond seen chains of thought for language models of sufficient scale.

# 6 Discussion

We have explored chain-of-thought prompting as a simple mechanism for eliciting multi-step reasoning behavior in large language models. We first saw that chain-of-thought prompting improves performance by a large margin on arithmetic reasoning, yielding improvements that are much stronger than ablations and robust to different annotators, exemplars, and language models (Section 3). Next,

---

[3]We tested 10 common names using GPT-3 `davinci` and it got all but one correct.

[4]For names of length longer than 2 words, we concatenate multiple first and last names together.

experiments on commonsense reasoning underscored how the linguistic nature of chain-of-thought reasoning makes it generally applicable (Section 4). Finally, we showed that for symbolic reasoning, chain-of-thought prompting facilitates OOD generalization to longer sequence lengths (Section 5). In all experiments, chain-of-thought reasoning is elicited simply by prompting an off-the-shelf language model. No language models were finetuned in the process of writing this paper.

The emergence of chain-of-thought reasoning as a result of model scale has been a prevailing theme (Wei et al., 2022b). For many reasoning tasks where standard prompting has a flat scaling curve, chain-of-thought prompting leads to dramatically increasing scaling curves. Chain-of-thought prompting appears to expand the set of tasks that large language models can perform successfully—in other words, our work underscores that standard prompting only provides a lower bound on the capabilities of large language models. This observation likely raises more questions than it answers—for instance, how much more can we expect reasoning ability to improve with a further increase in model scale? What other prompting methods might expand the range of tasks that language models can solve?

As for limitations, we first qualify that although chain of thought emulates the thought processes of human reasoners, this does not answer whether the neural network is actually "reasoning," which we leave as an open question. Second, although the cost of manually augmenting exemplars with chains of thought is minimal in the few-shot setting, such annotation costs could be prohibitive for finetuning (though this could potentially be surmounted with synthetic data generation, or zero-shot generalization). Third, there is no guarantee of correct reasoning paths, which can lead to both correct and incorrect answers; improving factual generations of language models is an open direction for future work (Rashkin et al., 2021; Ye and Durrett, 2022; Wiegreffe et al., 2022, *inter alia*). Finally, the emergence of chain-of-thought reasoning only at large model scales makes it costly to serve in real-world applications; further research could explore how to induce reasoning in smaller models.

# 7   Related Work

This work is inspired by many research areas, which we detail in an extended related work section (Appendix C). Here we describe two directions and associated papers that are perhaps most relevant.

The first relevant direction is using intermediate steps to solve reasoning problems. Ling et al. (2017) pioneer the idea of using natural language rationales to solve math word problems through a series of intermediate steps. Their work is a remarkable contrast to the literature using formal languages to reason (Roy et al., 2015; Chiang and Chen, 2019; Amini et al., 2019; Chen et al., 2019). Cobbe et al. (2021) extend Ling et al. (2017) by creating a larger dataset and using it to finetune a pretrained language model rather than training a model from scratch. In the domain of program synthesis, Nye et al. (2021) leverage language models to predict the final outputs of Python programs via first line-to-line predicting the intermediate computational results, and show that their step-by-step prediction method performs better than directly predicting the final outputs.

Naturally, this paper also relates closely to the large body of recent work on prompting. Since the popularization of few-shot prompting as given by Brown et al. (2020), several general approaches have improved the prompting ability of models, such as automatically learning prompts (Lester et al., 2021) or giving models instructions describing a task (Wei et al., 2022a; Sanh et al., 2022; Ouyang et al., 2022). Whereas these approaches improve or augment the input part of the prompt (e.g., instructions that are prepended to inputs), our work takes the orthogonal direction of augmenting the outputs of language models with a chain of thought.

# 8   Conclusions

We have explored chain-of-thought prompting as a simple and broadly applicable method for enhancing reasoning in language models. Through experiments on arithmetic, symbolic, and commonsense reasoning, we find that chain-of-thought reasoning is an emergent property of model scale that allows sufficiently large language models to perform reasoning tasks that otherwise have flat scaling curves. Broadening the range of reasoning tasks that language models can perform will hopefully inspire further work on language-based approaches to reasoning.

## Acknowledgements

We thank Jacob Devlin, Claire Cui, Andrew Dai, and Ellie Pavlick for providing feedback on the paper. We thank Jacob Austin, Yuhuai Wu, Henryk Michalewski, Aitor Lewkowycz, Charles Sutton, and Aakanksha Chowdhery for helpful discussions. We thank Sid Maxwell for notifying us about a mistake in the manual error analysis in the original manuscript.

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
