# OpenReview forum: "Chain-of-Thought Prompting Elicits Reasoning in Large Language Models"
_NeurIPS.cc/2022/Conference — NeurIPS 2022 Accept_

### Official Review · Reviewer_UpwE · 2022-07-10

**Rating:** 9
**Confidence:** 4
**Soundness:** 4 excellent
**Presentation:** 4 excellent
**Contribution:** 4 excellent

**Summary:**

This paper proposed chain of thought prompting, which can significantly improve the ability of large language models to perform complex reasoning. The proposed method is simple and effective, and also can generalize to various arithmetic, commonsense, and symbolic reasoning tasks.

**Questions:**

n/a

**Ethics Review Area:**

["I don’t know"]

**Strengths And Weaknesses:**

The paper is well motivated -- scaling up model size alone has not proved sufficient for achieving high performance on challenging tasks such as arithmetic, commonsense, and symbolic reasoning.

The proposed method – the chain of thought prompting – is novel, simple, and effective. The prompts help large language models to generate intermediate natural language reasoning steps that lead to the final output.

The experiments demonstrated the chain of thought prompting could generalize to various arithmetic, commonsense, and symbolic reasoning tasks.

Overall, there is no apparent weaknesses found in the paper. I champion for accepting this paper.

---

> ### Author Response · Authors · 2022-08-01
> **Response to review UpwE**
>
> Thank you for the positive review! We aimed to provide a comprehensive empirical evaluation of chain of thought prompting and we are very encouraged that you found the method to be simple and effective.

---

### Official Review · Reviewer_nMs2 · 2022-07-12

**Rating:** 6
**Confidence:** 4
**Soundness:** 4 excellent
**Presentation:** 4 excellent
**Contribution:** 3 good

**Summary:**

The paper proposes chain-of-thought prompts for in-context few-shot learning. The authors found that with properly designed prompts, which are decomposed reasoning sub-procedures, large language models can achieve substantially higher performances on the arithmetic reasoning, commonsense reasoning, and symbolic reasoning tasks that are tested in the experiments. For many tasks studied in this paper, the performance exceeds that of the supervised finetuning methods.  Analyses show that the chain-of-thought prompts significantly improve few-shot performance compared to the standard prompts. The performance of chain-of-thought prompts has been shown to be robust in different perspectives (e.g., different annotators and various language models).

**Questions:**

-	When annotator A, B, and C wrote chain of thought, what’re their strategies?

-	The paper mentioned that one contribution of the proposed model is facilitating interpretation and debugging of the few-shot learning models. Can the authors add some discussion on how to perform the debug?



**Limitations:**

please refer to the weakness comments.

**Strengths And Weaknesses:**

Strength:

-	The proposed model that incorporates chain of thought into prompts achieved substantial and consistent improvements on different tasks with different language models. Although the originality of the paper is a bit of concern, I do think these results are valuable to the community.
-	The paper performs comprehensive and detailed analysis to help establish the claims and draw the conclusions with solid evidence and justification. The detailed analyses provide insights and interesting discussions on the property of the model, e.g., the property with regard to language model sizes and complexities of the target problems, which may invite more studies along the line (although unfortunately, many of such studies may only be conducted in a limited number of organizations).
-	The paper is clearly written and I like the detailed discussions in the FAQ session, which further addressed my questions.

Weakness:

-	The originality and novelty, particularly from the methodology perspective, is limited. (However, due the above strengths, I would think the paper’s overall contributions outweighs this limitation.)
-	In terms of generalizability, the paper claims that the proposed approach is applicable in principle to any tasks that humans can solve via language. This is not well supported in the paper and may need further justification. (The contributions of the paper, though, are not dependent on that claim.). For example, the paper mentioned that spurious paths (wrong reasoning steps leading to correct answers) seem to be less of a concern in some of the datasets used in this paper, but that problem could be more serious in other complicated reasoning tasks. One may think that the reasoning problems that need more expressive/powerful model for commonsense may put challenges to the proposed approach.

---

> ### Author Response · Authors · 2022-08-01
> **Response to review nMs2**
>
> Thank you for the insightful review! Our paper explored a simple approach for improving reasoning in language models, and we are happy that your review found the improvements substantial and the analyses comprehensive (and that you liked the FAQ section). Below we answer the questions and describe how we have revised the paper.
>
> > When annotator A, B, and C wrote chain of thought, what’re their strategies?
>
> Thanks for asking about this detail. The annotators were not given specific instructions other than to simply write the step-by-step reasoning that led to the final answer. Thus, the annotations had each annotator’s own linguistic “chain of thought” writing style. This chain of thought annotation strategy is similar to the one in Cobbe et al., 2021. To make this more clear, in the revised paper we added in this detail in Appendix A.2.
>
> > The paper mentioned that one contribution of the proposed model is facilitating interpretation and debugging of the few-shot learning models. Can the authors add some discussion on how to perform the debug?
>
> Thank you for this question. A simple example of analyzing incorrect chains of thought could be to see what percent of errors involved calculation errors, where the language model correctly reasoned about the problem but made a simple calculator error (e.g., 6 * 13 = 68). This is described in detail in Appendix D.2. Based on manual analysis of reading chain of thought generations, we found that 8% of a subset of 50 errors from LM-1 137B on GSM8K involved such calculator errors. Accordingly with this observation, we implemented an ad-hoc calculator that uses python to parse and correct such calculation errors. This improved the accuracy of LM-1 by several points from 14.3% to 17.3%.
>
> > For example, the paper mentioned that spurious paths (wrong reasoning steps leading to correct answers) seem to be less of a concern in some of the datasets used in this paper, but that problem could be more serious in other complicated reasoning tasks. One may think that the reasoning problems that need more expressive/powerful model for commonsense may put challenges to the proposed approach.
>
> Thank you for this insightful observation. Indeed, not all model-generated reasoning paths are 100% perfect, and wrong reasoning can lead to both wrong answers and accidentally correct answers (which are both problematic). This is a general problem not just for the proposed method, but also more broadly in language model generation and neuro-symbolic reasoning (Maynez et al., 2021; Rashkin et al., 2021; Chen et al., 2019, and others). In the revised paper, we added this discussion into limitations (Section 6 paragraph 3), as well as in a new paragraph in Appendix D.2 (page 27). Overall, improving the factuality of generated reasoning paths is an important open direction for future research. For instance, training a re-ranker to score many randomly-sampled generations is a potential technique for improving the quality of language model generations (Shen et al., 2021; Thoppilan et al., 2021; inter alia).
>
> > In terms of generalizability, the paper claims that the proposed approach is applicable in principle to any tasks that humans can solve via language. This is not well supported in the paper and may need further justification. (The contributions of the paper, though, are not dependent on that claim.)
>
> Thank you for pointing this out (and noting that the paper’s contributions are not dependent on this claim). While the paper focused on 12 reasoning benchmarks, it is hard to empirically show that chain of thought prompting works for all language tasks (hence why it is caveated with “in principle”). We have softened the language in the revised paper (added the word “potentially” in addition to “in principle” in Sections 1, 2, and 7). In addition, we added an additional paragraph in A.3 that explicitly says that further empirical experiments are needed to evaluate chain of thought prompting on non-reasoning tasks (e.g., machine translation), and that we leave it for future work. Please let us know if you have any suggestions for further clarifying this!

---

### Official Review · Reviewer_TQF2 · 2022-07-16

**Rating:** 6
**Confidence:** 3
**Soundness:** 3 good
**Presentation:** 3 good
**Contribution:** 3 good

**Summary:**

This paper introduces Chain of Thought (CoT) prompting, which augments few-shot training samples for in-context learning with explicit reasoning steps. They evaluate the effectiveness of CoT on math word problems, commonsense reasoning and symbolic reasoning and shows performance improvements from the standard prompting as well as fine-tuned GPT-3 with smaller numner of parameters. In addition, a 540B parameter language model prompted with CoT outperforms fine-tuned 175B GPT-3.

**Questions:**

- Is it possible to test CoT with the publiclly available models with the same scale (e.g., Meta OPT)?

**Limitations:**

The authors mention several limitations of CoT (e.g., writing CoT prompts).

**Strengths And Weaknesses:**

## Strengths
- The proposed idea is easy, simple, (potentially) widely-applicable and well-motivated.
- They show that CoT gives large performance improvements when the LMs are large enough, although performance improvements are somewheat limited when the base LMs are smaller than 100B.
- Their through analysis shows that (i) in many cases the models' generated explanations are correct, (ii) removing sub part of CoT or changing the answer / CoT order largely degrades models' performance, indicating the necessity of using the reasoning process writte in natural languages, and (iii) CoT is robust to the different expalanation written by different annotations (Author A, B and C).

## Weaknesses
- Although the performance improvements given by CoT is impressive when the LMs are big enough, with the models smaller than 175B, CoT prompting does not provide sigifincat improvements from the standard prompting. Many groups cannot afford to use 175B or larger models, which may make the aplicability and usefulness of CoT limited. Moreover, the evaluations are done in mostly relatively limited domains (e.g., mathword problem). Recent work (Ye and Durret, 2022)* shows that explanations generated by GPT-3 may not even be factually grounded in the input, even on simple tasks with straightforward extractive explanations.
- All of the models used in this papers are only particially publically available (GPT-3 APIs) or completley publically unavailable (LM1 and LM2), which makes it difficult to verify the experimental results and findings in this paper.

=======
[2022/08/08] I updated my score given the new results and dicussions with the authors.

*Ye, Xi, and Greg Durrett. "The Unreliability of Explanations in Few-Shot In-Context Learning." arXiv preprint arXiv:2205.03401 (2022).

---

> ### Author Response · Authors · 2022-08-01
> **Response to review TQF2**
>
> Thank you for the detailed review! Our paper evaluated a simple method for improving the reasoning abilities of language models, and we are glad you found it well-motivated and thorough. Based on your suggestions, we have conducted new experiments and added them to the revised paper.
>
> > Although the performance improvements by CoT is impressive when the LMs are big, with models smaller than 175B, CoT prompting does not provide significant improvements from standard prompting. Many groups cannot afford 175B or larger models, which may make the applicability and usefulness of CoT limited.
>
> Thank you for noting that CoT prompting is an “emergent ability” of scaling. Doing experiments on the GPT-3 API, though publicly available, costs money. For this reason, we further added evaluation on Codex (https://openai.com/blog/openai-codex/), which is currently free to use via the OpenAI API.
>
> Similar to the three models already in the paper, chain of thought prompting substantially outperforms the baseline when using Codex. We added these results to the revised paper (Appendix B, Tables 1-4) and summarize them here:
>
> | Codex Results                     | GSM8K | SVAMP | ASDiv | AQuA | MAWPS | Avg.         |
> |-----------------------------------|-------|-------|-------|------|-------|--------------|
> | Standard prompting                | 19.7  | 69.9  | 74.0  | 29.5 | 78.7  | 54.4         |
> | Chain of thought prompting        | 63.1  | 76.4  | 80.4  | 45.3 | 92.6  | 71.6 **(+17.2)** |
> | Chain of thought prompting + calc | 65.4  | 77.0  | 80.0  | 45.3 | 93.3  | 72.2 **(+17.9)** |
>
> | Codex Results              | CSQA | StrategyQA | Date | Sports | SayCan | Avg.        |
> |----------------------------|------|------------|------|--------|--------|-------------|
> | Standard prompting         | 82.3 | 67.1       | 49.0 | 71.7   | 85.8   | 71.2        |
> | Chain of thought prompting | 77.9 | 73.2       | 64.8 | 98.5   | 88.3   | 80.5 **(+9.3)** |
>
> We really appreciate this suggestion, because the Codex results are comparable to or better than the 540B model we evaluate. Notably, Codex does well on both arithmetic and commonsense reasoning tasks. As Codex is free, anyone can reproduce this result.
>
> > All of the models … are only partially publically available (GPT-3 APIs) or completely publically unavailable (LM1 and LM2), which makes it difficult to verify the experimental results. Is it possible to test CoT with the publically available models with the same scale (e.g., Meta OPT)?
>
> In addition to the three large language models we tested in the paper and the Codex result above, we further did an additional evaluation using UL2 20B (https://arxiv.org/abs/2205.05131).
>
> UL2 is a new, completely open-source model that combines various language learning objectives. We chose UL2 because it has similar or better performance than GPT-3 and Meta OPT on zero-shot SuperGLUE, and it is only 20B parameters (compared to 175B parameters in OPT). Thus, the publicly-available UL2 is more compute-friendly than OPT, in addition to being higher-performance.
>
> We added these results to the paper (Appendix B, Tables 1-4), and also show them in the below tables. For nine out of ten arithmetic/commonsense tasks, CoT prompting improves performance compared to standard prompting. The UL2 results are weaker in raw performance than the 540B and InstructGPT models (see Tables 1–5), but they are comparable to LM-1 68B, which is good considering that UL2 is only 20B. With an external calculator (implementation described in Appendix B), the average gain on arithmetic reasoning jumps to (13.7%), which suggests that many of UL2’s arithmetic errors are from calculations instead of reasoning (potentially because C4 is a noisy pre-training dataset). Note that calculator is not applicable for standard prompting since there are no intermediate reasoning paths.
>
>
> | UL2 Results                       | GSM8K | SVAMP | ASDiv | AQuA | MAWPS | Avg           |
> |-----------------------------------|-------|-------|-------|------|-------|---------------|
> | Standard prompting                | 4.1   | 10.1  | 16.0  | 20.5 | 16.6  | 13.5          |
> | Chain of thought prompting        | 4.4   | 12.5  | 16.9  | 23.6 | 19.1  | 15.3 **(+1.8%)**  |
> | Chain of thought prompting + calc | 6.9   | 28.3  | 34.3  | 23.6 | 42.7  | 27.2 **(+13.7%)** |
>
> | UL2 Results                | CSQA | StrategyQA | Date | Sports | SayCan | Avg          |
> |----------------------------|------|------------|------|--------|--------|--------------|
> | Standard prompting         | 34.2 | 59.0       | 13.5 | 57.9   | 20.0   | 36.9         |
> | Chain of thought prompting | 51.4 | 53.3       | 14.0 | 65.3   | 41.7   | 45.1 **(+8.2%)** |
>
> One final note is while CoT currently works best with large models, there is growing research making language models more computationally efficient (Chinchilla), and more accessible (BLOOM). Hence, such methods may gain even further traction and are forward-looking.

---

> > ### Author Response · Authors · 2022-08-01
> > **Response continued**
> >
> > > Recent work (Ye and Durret, 2022) shows that explanations generated by GPT-3 may not even be factually grounded in the input, even on simple tasks with straightforward extractive explanations.
> >
> > Thank you for this pointer. Yes, one of the most important open problems in language model and natural language generation research broadly is generating factual outputs that are grounded on the input context (Maynez et al., 2021; Rashkin et al., 2021; Ye and Durret, 2022; Marasovic et al., 2022; Weigreffe et al., 2022; and more). The fact that language models (even with CoT prompting) do not solve elementary-school level math problems with perfect accuracy (100%) is consistent with the observation that language model generations can have factual hallucinations that lead to incorrect reasoning. We acknowledge that there is further room for factuality improvements, which could in turn improve performance.
> >
> > In the revised paper, we added additional discussion of this, both as a limitation in Section 6 Paragraph 3 and a new paragraph in Appendix D.2. We leave performance and factuality improvements for future work (and indeed, Ye and Durret 2022 actually show that such explanations, even if unreliable, can be used calibrate model predictions and hence improve performance).
> >
> > > The evaluations are done in mostly relatively limited domains (e.g., mathword problem)
> >
> > Thanks for this comment. The paper evaluated chain of thought prompting on five math word problem benchmarks. However, it also includes experiments on two symbolic reasoning datasets (last letter concatenation, coin flip), as well as five commonsense reasoning benchmarks (CommonsenseQA, StrategyQA, Date Understanding, and Sports Understanding, and SayCan). This covers a broad range of reasoning tasks. We leave evaluation on other types of tasks (e.g., traditional NLP tasks such as machine translation) for future work. We have further clarified this in Appendix A.3.

---

> > > ### Comment · Reviewer_TQF2 · 2022-08-05
> > > **Thank you for all of the new results! I have a few follow up questions.**
> > >
> > > Thank you for adding all of the new results and adding new discussions. The results using other models also look promising.
> > >
> > > > We added these results to the paper (Appendix B, Tables 1-4), and also show them in the below tables. For nine out of ten arithmetic/commonsense tasks, CoT prompting improves performance compared to standard prompting.
> > >
> > > The results are interesting. On Codex or your LM2 CoT generally gives large performance improvements across different datasets while with UL2, the results are mixed.
> > >
> > > > With an external calculator (implementation described in Appendix B), the average gain on arithmetic reasoning jumps to (13.7%), which suggests that many of UL2’s arithmetic errors are from calculations instead of reasoning (potentially because C4 is a noisy pre-training dataset).
> > >
> > > I agree with the point that the model may make more calculation errors, but on AQuA, adding the calculator doesn't help, which mght indicate that there's another issue besides the caldulation error. Do you have any inights into why the performance improvement by CoT with smaller models is somewheat limted or smaller than the ones with larger LMs?
> > >
> > > Also on non arithmatic tasks particularily in StrategyQA, adding CoT actually hurts the performance. Do you have any thoughts on why the model performance deteriorates on StrategyQA with CoT?

---

> > > > ### Author Response · Authors · 2022-08-06
> > > > **Thanks for the discussion!**
> > > >
> > > > Thanks for the further discussion!
> > > >
> > > > > Do you have any insights into why the performance improvement by CoT with smaller models is somewheat limited or smaller than the ones with larger LMs?
> > > >
> > > > Our thoughts on this are detailed in the FAQ on page 16, Question 1 (Appendix A.1, and the corresponding Figures 9 and 10), and we summarize them below:
> > > > - Why model scale improves chain of thought prompting is a very interesting open question. Chain of thought prompting is just one of many emergent abilities that have been observed in recent years that substantially improves as a result of scaling.
> > > > - The answer to this question will likely be multi-faceted, and we consider it mostly out of the scope of this current paper. Nonetheless, we did a small preliminary error analysis.
> > > >     - We manually examined 45 problems that a 62B model got wrong and looked at what percentage of them were fixed by scaling to 540B.
> > > >     - These errors were broadly grouped into semantic understanding, one-step missing, and other (hallucinations, repetitive outputs, symbol mapping errors).
> > > >     - For all three of these categories, the 540B model got a substantial portion of these questions correct. This suggests that some skills that the 62B model may lack are achieved by scaling up to 540B, which is consistent with a hypothesis that language models acquire a range of semantic understanding and logical reasoning skills as a function of scale.
> > > > - This observation that language models acquire many abilities at scale is consistent with a lot of prior literature (see the GPT-3 paper—Brown et al., 2020; the BIG-Bench paper—Srivastava et al., 2022, and many more).
> > > > - In summary, one hypothesis is that successful multi-step reasoning requires a combination of many skills (semantic understanding, symbol mapping, coherent text generation, staying on topic, etc) that likely improve as a function of model scale.
> > > >
> > > > > Do you have any thoughts on why the model performance deteriorates on StrategyQA with CoT?
> > > >
> > > > Thanks for this question. One reason why small models have worse CoT performance on StrategyQA is that StrategyQA requires substantial world knowledge (e.g., an example question from StrategyQA is *“Is growing seedless cucumber good for a gardener with entomophobia?”*). Smaller models have fewer parameters to memorize such world knowledge (e.g., Roberts et al., 2020), and so they are less likely to be able to produce correct reasoning steps, thus deteriorating performance.
> > > >
> > > > > on AQuA, adding the calculator doesn't help
> > > >
> > > > Yes, thanks for pointing this out. Unlike all the other datasets which are free response, AQuA is a multiple choice dataset (see Table 21 for an example). Our external calculation function only applied for free response questions since it did not have functionality to change the mappings to final multiple choice answers. We will clarify this in the next version.
> > > >
> > > > If you have more questions we will be happy to discuss further! We hope that these new experimental results addressed the concerns in your review feedback.
> > > >
> > > > References in this response:
> > > > - Adam Roberts, Colin Raffel, and Noam Shazeer. How much knowledge can you pack into the parameters of a language model? EMNLP 2020. https://arxiv.org/abs/2002.08910
> > > > - Srivastava et al., and 300+ authors. Beyond the Imitation Game: Quantifying and extrapolating the capabilities of language models. https://arxiv.org/abs/2206.04615

---

> > > > > ### Comment · Reviewer_TQF2 · 2022-08-08
> > > > > **Thank you for answering questions! I'm happy to increase my raiting.**
> > > > >
> > > > > Thank you for addressing my concersn and answering my quesitons! The experimental results and discussions are intriguing. I'm happy to increase my score.

---

### Author Response · Authors · 2022-08-01
**Overall comment**

Thank you all for the detailed reviews! Our paper explored a simple approach for multi-step reasoning via language models. In an empirical evaluation on twelve reasoning benchmarks, the approach improved performance over standard prompting by a large margin, achieving SOTA results on several benchmarks. We are encouraged that the reviewers agreed on the strengths of the work, including that the method is “simple and well-motivated” (TQF2, UpwE), the analysis is “comprehensive/thorough” (nMs2, TQF2), and the empirical gains are “large/substantial/effective” (all three reviewers).

In the individual replies, we describe additional improvements that we have made based on reviewer suggestions, which we believe make the paper stronger. Notably, we did new evaluations on UL2 and Codex, for which chain of thought prompting substantially outperformed standard prompting (consistent with the three models already evaluated in the paper). We hope this addresses the comment raised by Reviewer TQF2, as UL2 is a completely open-source model and Codex is currently free (and the strongest overall model we evaluated), which facilitates reproducibility.

We enjoyed discussing the feedback. Should the reviewers decide to engage further, we are happy to continue!

---

### Meta-Review · Area_Chair_uSET · 2022-08-24

**Recommendation:** Accept
**Confidence:** Certain

**Metareview:**

All reviewers have voted to accept the paper, and this is a solid work.

**Award:**

No

---

### Decision · Program_Chairs · 2022-09-14

Accept